# USCILab3D: A Large-scale, Long-term, Semantically Annotated Outdoor Dataset

**Kiran Lekkala**,* **Henghui Bao**\*, **Peixu Cai, Wei Zer Lim, Chen Liu, Laurent Itti**
University of Southern California
Los Angeles, CA 90089, USA
`klekkala@usc.edu`

## Abstract

In this paper, we introduce the **USCILab3D dataset**, a large-scale, annotated outdoor dataset designed for versatile applications across multiple domains, including computer vision, robotics, and machine learning. The dataset was acquired using a mobile robot equipped with 5 cameras and a 32-beam, $360°$ scanning LIDAR. The robot was teleoperated, over the course of a year and under a variety of weather and lighting conditions, through a rich variety of paths within the USC campus (229 acres = $\sim$ 92.7 hectares). The raw data was annotated using state-of-the-art large foundation models, and processed to provide multi-view imagery, 3D reconstructions, semantically-annotated images and point clouds (267 semantic categories), and text descriptions of images and objects within. The dataset also offers a diverse array of complex analyses using pose-stamping and trajectory data. In sum, the dataset offers 1.4M point clouds and 10M images ($\sim$ 6TB of data). Despite covering a narrower geographical scope compared to a whole-city dataset, our dataset prioritizes intricate intersections along with denser multi-view scene images and semantic point clouds, enabling more precise 3D labelling and facilitating a broader spectrum of 3D vision tasks. For data, code and more details, please visit our website.

## 1 Introduction

With the recent advancements in 3D vision techniques, the integration of three-dimensional perception has become integral to many interdisciplinary domains. Unlike the abundant resources available for 2D vision, the lack of comprehensive datasets for 3D vision poses a significant challenge to researchers. The progress in this field can be significantly propelled by leveraging large-scale datasets, which offer adaptability across a spectrum of downstream tasks.

In this paper, we present USCILab3D — a large-scale, long-term, semantically annotated outdoor dataset. USCILab3D comprises over 10 million images and 1.4 million semantic point clouds, rendering it suitable for a wide range of vision tasks.

Differing from smaller-scale semantic datasets or larger-scale undetailed ones, our dataset not only encompasses a wide array of outdoor multi-view scene images but also provides detailed semantic annotations, facilitating enhanced understanding and utilization of 3D perception techniques. Given the massive scale of our new dataset, as detailed below, we have thus far focused on leveraging the latest foundation models to compute detailed annotations. Our workflow using these models is detailed below.

---

*Equal Contribution.

38th Conference on Neural Information Processing Systems (NeurIPS 2024) Track on Datasets and Benchmarks.

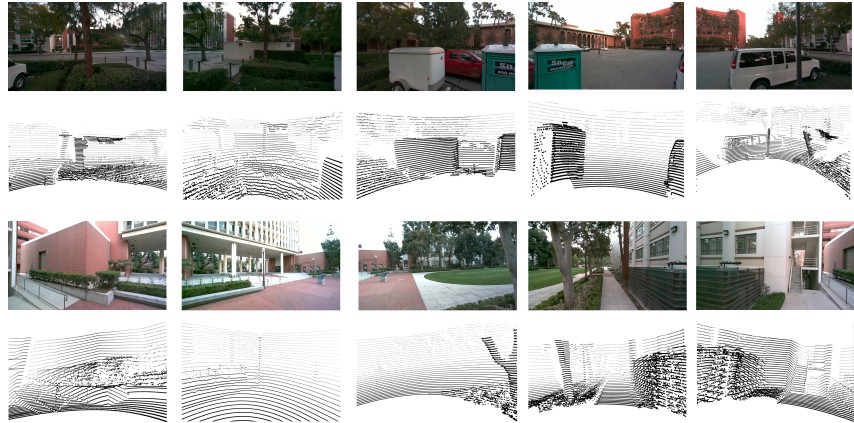

Figure 1: **Images with the respective 3D pointclouds** Our adjacent five cameras provide comprehensive coverage with overlap at the same timeframe, ensuring the captured information's redundancy. We also show the corresponding point cloud view for every image.

## 2  Related datasets

Several large-scale scene datasets have been developed in recent years for indoor settings [19; 26; 21]. Additionally, several datasets have focused on outdoor city navigation[18]. Furthermore, some datasets are generated using simulators [9; 24]. These attempt to solve the above problems, although presenting their challenges: While they offer controlled environments, there exists a noticeable gap in scene quality compared to real-world scenes.

### 2.1  Multi-view datasets

Multi-view scene datasets are typically used for novel view synthesis tasks with generative models such as Neural Radiance Fields (NeRF) [17] and 3D Gaussian Splatting [14]. The LLFF dataset [16] is an early multi-view scene dataset that includes both indoor and outdoor scenes, with fewer than 1,000 low-resolution images. The DTU [13] and ScanNet [8] datasets contain between 30K and 2,500K images, but they are limited to indoor scenes. The ETH3D dataset [23] provides high-quality outdoor scenes but has sparse scans and fewer than 1,000 images. Tanks and Temples [15] addresses these limitations by offering 147,000 high-quality outdoor images, which are commonly used in novel view synthesis benchmarks.

### 2.2  Scene datasets with semantic labels

**Indoor datasets** Datasets like [19; 26] represent large-scale 3D reconstruction datasets tailored for research in indoor robotic navigation and scene understanding. Matterport [6] is a large-scale RGB-D indoor dataset containing 10,800 panoramic views from 194,400 RGB-D images of 90 building-scale scenes. However, this dataset is limited to indoor environments and offers only 20 labels for scene annotation. In contrast, our dataset encompasses approximately 10 million images and over 4000 labels, providing extensive coverage of outdoor scenes. Moreover, the inclusion of ground-truth point clouds in our dataset enhances the accuracy of alignment between 2D images and 3D annotations, surpassing the alignment capabilities of other datasets.

**Outdoor datasets** SemanticKITTI [4] is a widely used dataset for semantic segmentation and scene understanding in outdoor environments. It consists of dense point cloud sequences collected by a mobile LiDAR scanner which is similar to us. However, SemanticKITTI's semantic annotations are confined to only 25 categories. In contrast, leveraging multimodal model outputs, our dataset enables the labeling of almost every element within the scene, providing a comprehensive understanding of outdoor environments.

Our dataset addresses the limitations of the above datasets by providing large-scale outdoor scenes with diverse weather and lighting conditions, along with various ground-truth semantic point clouds (Table 1 and Table 2). Leveraging multimodal foundational models, we accurately label 2D images

and align them in 3D space, resulting in precise 3D annotations.

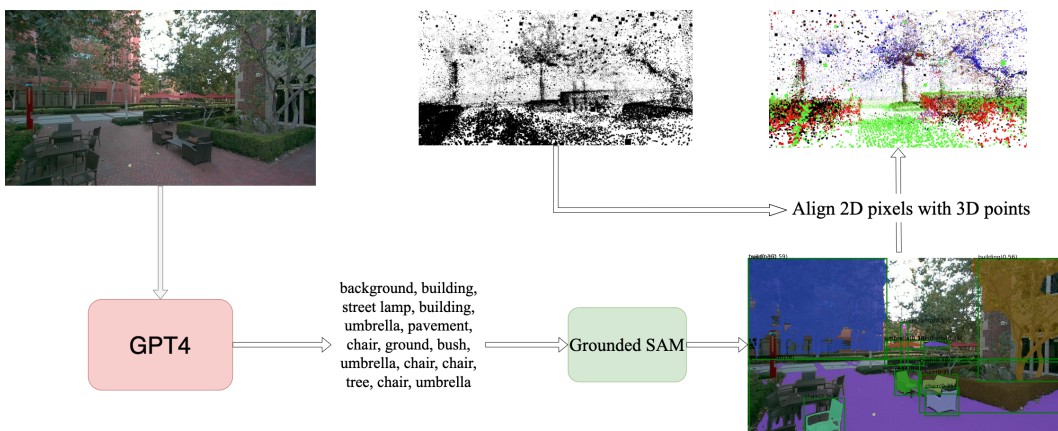

Figure 2: **The pipeline of our semantic annotations method** We use GPT4 and Grounded-SAM to create pixel-wise semantic labels and align the 2D and 3D points.

| Dataset | Frames | Indoor | Outdoor | LiDAR Point Cloud | Semantic |
|---|---|---|---|---|---|
| LLFF[16] | < 1K images | ✓ | ✓ | ✗ | ✗ |
| DTU[13] | 30K images | ✓ | ✗ | ✗ | ✗ |
| ScanNet[8] | 2,500K images | ✓ | ✗ | ✗ | ✗ |
| Tanks and Temples[15] | 147K images | ✓ | ✓ | ✗ | ✗ |
| ETH3D[23] | <1K images | ✓ | ✗ | ✗ | ✗ |
| Matterport3D[6] | 195K images | ✓ | ✗ | ✗ | ✓ |
| Habitat[19] | - | ✓ | ✗ | ✗ | ✓ |
| iGibson[26] | - | ✓ | ✗ | ✓ | ✓ |
| SemanticKITTI[4] | 23K scans | ✗ | ✓ | ✓ | ✓ |
| USCILab3d (ours) | 10M images 1.4M scans | ✗ | ✓ | ✓ | ✓ |

Table 1: Comparison of the existing datasets with our USCILab3D dataset.

| Dataset | Point Clouds | Semantic Labels | Semantic classes |
|---|---|---|---|
| nuScenes[5] | 390K | 31 | vehicle, human, animal, movable object, flat, static |
| Waymo motion[11] | 230K | 23 | Traffic Entities: Car, Truck, Bus, Motorcyclist, Bicyclist, Pedestrian, etc. |
| SemanticSTF[29] | 2K | 21 | flat, construction, nature, vehicle, human, object |
| WildScenes[28] | 12K | 15 | terrain, vegetation, object, structure, water, sky |
| **USCILab3d (Ours)** | 1.4M | 267 | Vehicle, nature, human, ground, structure, street furniture, architectural elements, signs and symbols, general objects, lightning |

Table 2: Comparison of Semantic Classes and Labels Across Existing Datasets and Our USCILab3D Dataset.

# 3 Dataset collection

This section outlines our robot platform and data collection approach. Our robot, Beobot-v3, utilizes multiple cameras and a LiDAR sensor for simultaneous data capture. We collect data across the USC University Park campus and synchronize streams for analysis.

## 3.1 Robot platform

We build our robot Beobot-v3 to collect the dataset, as shown in Figure 3. We use five Intel Realsense D455 cameras and Velodyne HDL-32E LiDAR. The RGB images, featuring a field of view (FOV) of $90 \times 65°$ and a resolution of $1280 \times 720$ pixels, are captured at a rate of 15 frames per second (FPS).

Utilizing a 1 MP RGB sensor, these images ensure high-quality visual data acquisition. Furthermore, the LiDAR scans the environment at a rate of 10 Hz, capturing precise point clouds that complement the visual data. These point clouds offer comprehensive 3D spatial information essential for scene understanding and navigation tasks. Because of microcomputer's limit, camera 1 and LiDAR are controlled by one microcomputer, and other cameras are controlled by their own microcomputer. All microcomputers are controlled by a central computer, our data collection system orchestrates the simultaneous scanning and recording process. As the LiDAR initiates scanning, capturing a 360° view of the environment, the data is saved directly into the system and five cameras capture images in tandem, storing them in separate ROS bag files.

## 3.2 Dataset collected over the entire USC campus

Our dataset is meticulously collected across the entirety of the USC University Park campus. Spanning an expansive area of 229 acres (0.93 km²), the campus makes our dataset diverse. From the varied architecture of its buildings to the network of roads, stairs, trails, paths, gardens, and sidewalks, each corner offers a unique scene. By dynamically selecting its route, the robot explores the full extent of the campus' diverse terrain, from thoroughfares to hidden nooks, creating a rich variety of surroundings.

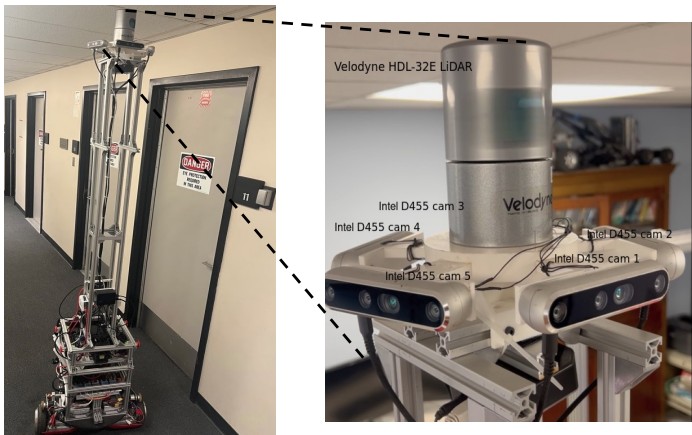

Figure 3: **Overview of the data collection robot and its hardware.** Beobot-v3 is a differential-drive, non-holonomic mobile robot, equipped with five Intel Realsense D455 cameras and one Velodyne HDL-32E LiDAR sensor used to collect the dataset.

The data collection occurred in many daytime sessions, with a preference for sunrise or sunset periods to avoid crowds and mitigate harsh sunlight that could degrade image quality. However, a small portion of the captured images may still exhibit the effects of powerful sunshine. The sample images are shown in Figure 4.

Our data collection efforts span from March 11, 2023, to March 16, 2024, encompassing 12 months. Over this time frame, the environment undergoes dynamic changes, including variations in weather, seasons, and alterations to the campus landscape, such as ongoing construction projects. This deliberate scheduling ensures that our dataset encapsulates a diverse range of environmental scenarios, enriching the dataset with a wide array of conditions for robust training and evaluation of algorithms.

## 3.3 Synchronization of cameras and LiDAR

To address the synchronization issue between the LiDAR and cameras due to the control of different microprocessors, we implement a synchronization process. Given that the LiDAR operates on the same system clock as camera 1, we only need to synchronize the remaining cameras with camera 1. To achieve this, we employ a method based on feature detection and optical flow tracking. At the onset of each session, the scene remains static. Leveraging ShiTomasi corner detection [27], we identify key features in the camera images. Subsequently, using the Lucas-Kanade optical flow algorithm, we track the movement of these features over consecutive frames. If the displacement

of these features exceeds a predefined threshold, indicative of the robot initiating movement, we designate this time as the session's start time.

Once the start time is determined for camera 1, we synchronize the start times of the remaining cameras by aligning them with the start time of camera 1. This ensures temporal coherence across all camera feeds, enabling accurate alignment of the visual and LiDAR data streams. Through this synchronization process, we establish temporal consistency across all data sources, facilitating coherent analysis and interpretation of the collected data.

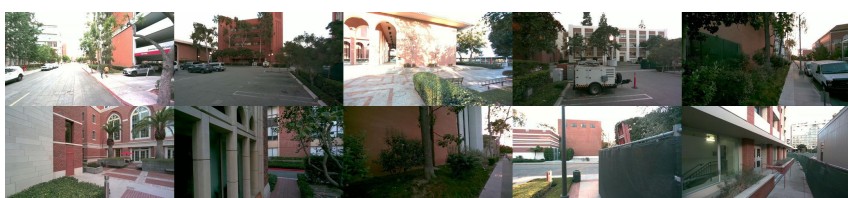

Figure 4: **Sample snapshots from our dataset of various daylight timings**. These are images obtained from randomly sampling across the entire dataset.

### 3.4  Sensor calibration

By aligning the coordinate systems of the Velodyne LiDAR and the camera, we ensure that the geometric transformation from 3D to 2D space is accurate. With this calibrated setup, we can assign semantic labels to the 3D points based on the information extracted from the images. The accurate alignment between the Velodyne-frame and camera-frame ensures that the projected points correspond to the correct regions in the images, enabling us to leverage the semantic information obtained from the images to label the 3D points accurately.

To obtain the pose transformation between images and point clouds, we use a 1m × 1m checkerboard as a calibration target for sensor alignment. Leveraging the MATLAB calibration toolbox, we apply the Line and Plane Correspondence method [30] to refine sensor alignment and calibration with high precision. In this approach, we treat edges in 3D as contours (C) and planes (a), while lines (L) in 3D space are characterized by points within the same plane (a). This framework integrates point-to-line, point-to-plane, and direction/normal-based adjustments, ensuring accurate alignment across sensors.

## 4  Dataset annotation

In this section, we describe methods used as part of the pipeline for our semantic annotations of 3D point clouds. A high-level overview is shown in Figure 2.

### 4.1  GPT4-based candidate labels and clustering

We use GPT-4 [1] to detect the semantic labels in an image. Since images are obtained at 15Hz and the robot moves at a velocity close to 1 m/s, it is redundant and expensive to query the semantic labels for all images through GPT-4 model. Given that the image frequency is 15Hz, for about every 225 images from one camera, we extract the the images of five cameras at that time. Given that the camera records at 15Hz, a 15-second interval of movement (typically less than 12 meters) ensures a small scene variation.

We then pass these 5 images to GPT-4, and prompt it to estimate the semantic labels of the images using the following prompt *"List every possible semantic class that exists in the scene. List only the names and nothing else."* After standardizing and filtering the output, we obtain a total of 4162 labels. But most labels are meaningless or have similar meaning. We then again use GPT-4 to perform clustering and categorization on the estimated semantic labels.

After removing the meaningless labels and merging semantically equivalent labels, we obtained 257 unique labels. Then, for all images we asked GPT-4 to extract objects from the image again, now with prompt is "I will give you a list of semantic class, list every possible semantic class that exists in

the scene. List only the names and nothing else, split by comma." This yields the final label list for each image.

| Category | Elements |
|---|---|
| Vehicle | vehicle, bicycle, van, truck, motorcycle, golf cart, bus, car, skateboard |
| Nature | sky, grass, tree, shrub, shrubbery, hedge, trunk, tree trunk, green area, birds, bush, yard, plant |
| | sun, palm, rock, soil, leaf, leaves, water, flower, branch, bushes, vegetation, bird, ivy |
| Human | person, hand |
| Ground | pavement, curb, gravel, rail, sidewalk, street, walkway, floor, road, pedestrian walkway, crosswalk |
| | ramp, garden, ground, pathway, paving stone, golf course, parking lot, drainage grate, mulch |
| Structure | monument, structure, courtyard, fountain, public space, construction, emergency station |
| | ceiling, fence, gate, wall, balcony, container, stadium, lattice, shed, house, construction |
| | pipe, roof, building, sports field, campus, toilet, baseball field, architecture |
| | site, parking structure, garage, scaffolding, archway, call station |
| Street Furniture | bench, pole, feeding station, patio, handicap, barrier, hydrant, construction cone, construction barrier |
| | lamp post, lamp, trash can, recept, sign, parking meter, public art, statue, sculpture |
| | bollard, bus stop, park bench |
| Architectural Elements | drain cover, manhole cover, vent, air vent, arch, sill, doorway, baluster, security camera, electric box |
| | corridor, stair, ventilation grill, door handle, entrance, post, air unit, pillar, balustrade, handrail |
| | window, door, elevator, gutter, bleachers, tank, generator, utility meter |
| General Objects | umbrella, table, chair, stroller, furniture, board, bottle, canopy, outdoor gear, advertisement, station |
| | pot, rack, flag, locker, ladder, garbage, bulletin board, pallet, planter, equipment, tent, base, hat |
| | curtain, blinds, cardboard, box, tire, wheels, bag, bed, frame, bucket, painting, poster , machine |
| Signs and Symbols | shadow, reflection, traffic cone, parking space line, space line, road marking |
| | parking symbol, stop sign, street sign, road sign, symbol, plaque, banner, graffiti, waste container |
| | signboard, security camera, camera, warning sign, fire safety sign, transportation sign |
| | handicap sign, closed sign, exit sign, parking sign, reservation sign, rec sign |
| Materials | concrete, brick, construction materials, stone, wood, plastic, metal, glass, iron, materials |
| Lighting | outdoor lighting, light, street light, indoor light, lantern, sunlight, shade |
| Miscellaneous | cover, trash, outdoor, chain, unit, security, exterior, fire, electric, meter, lettering, phone, debris, railway |
| | text, potted, space, portable, cone, stlight, cross, marker, grate, blea, stoller, units, picnic, electrical |
| | cable, basin, pavilion, ster, bal, field, curve, bod, bay, pal, firent, box, exit, baseball, image, rec, sports |
| | public, piping, grill, guttering, utility, call, case, recacle, gut, hydra, air |
| | line, tile, cardboard, patch, reservoir, valve |

Table 3: **Clustering of the semantic labels.** We use GPT-4 to cluster 267 labels into 12 categories using the prompt "Could you help me classify by following category: Vehicle, Nature, Human, Ground, Structure, Street Furniture, Architectural Elements."

## 4.2 Grounded-SAM masks on pixel space

After we obtain the candidate labels, for equally spaced subset of images, we use those labels as an input to the Grounded-SAM model [20] to detect and segment the image by pixel. Since we are using a differential-drive robot that can potentially rotate left or right, images may look very different quite rapidly, so we merge the five image labels from GPT-4 and pass to next step. After conducting our experiments, we found that the presence of unrelated labels (not visually represented in the images) does not significantly influence the results of Grounded-SAM. This observation is reflected in Figure 5 and Table 4 through the percentage of incorrect pixel labels in the masks of 2 images. We show the top 50 frequent objects and their pixel percentage in images of our dataset in Figure 6.

## 4.3 Post-processing after Grounded-SAM

Grounded-SAM's output is not always using the same vocabulary as our input labels, e.g., one may prompt it for 'vehicle' but obtain a segmented 'car'. It may also generate meaningless words or words having similar meaning. To address this, we perform clustering and categorization as in section 4.1 again to merge all similar labels. Additionally, we manually merge and remove some words. Ultimately, we obtain 267 labels and 12 categories (Table 3).

## 4.4 Projecting 2D semantic masks to 3D pointcloud

From the LIDAR data, we reconstruct 3D trajectories of the robot throughout the dataset. Essentially, we compute a pose transformation for each LiDAR scan in the dataset. We then interpolate the LiDAR poses to the camera images using the extrinsic parameters corresponding to the transformation of each camera with respect to the LiDAR sensor. This results in a pose estimate for every camera image in the dataset.

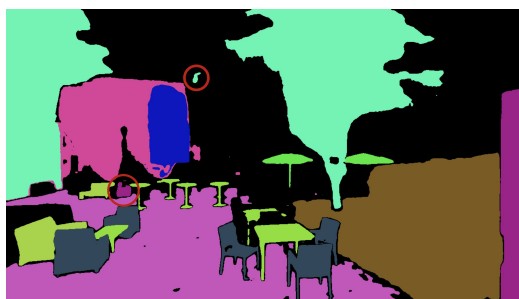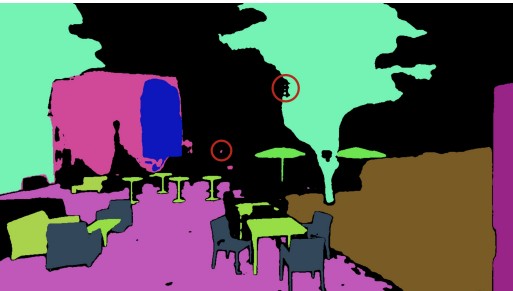

Figure 5: **Robustness of Grounded SAM to prompts.** Comparison of the semantic masks obtained using different prompts for the same image by Grounded-SAM model, showing the robustness of the model. On the right image, the additional prompts were *"fire hydrant, person, car, Parking lot lines, Boat, Scooter, Dog, Bear, Cat"* along with the common prompts *"Trees, Bushes, Benches, Tables, Chairs, Pavement, Buildings, Windows, Doors, Emergency call box, Umbrellas, Leaves, Grass"*

| Additional prompts | Incorrect pixel labels |
|:---:|:---:|
| 1 | 0.23% |
| 2 | 0.63% |
| 3 | 0.63% |
| 10 | 0.92% |

Table 4: **Percentage of incorrect pixel labels**. Quantitative measures to show robustness through the change in the percentage of incorrect pixel labels with additional prompts. Note that this table in relation with the above Figure 5

By utilizing the semantic map of every image obtained from Grounded SAM, we use ground truth camera intrinsics and extrinsics to accurately project 3D point clouds onto 2D images, following equation (1). Here, $(X, Y, Z)$ represents the world coordinates of a point, while $(x, y)$ denotes the coordinates of the point projected onto the image plane, measured in pixels. $r$ and $t$ are rotation and translation. $c_x, c_y$ represents the principal point, and $f_x, f_y$ are the focal lengths in pixels. Subsequently, we align the 2D and 3D points to assign labels to the 3D points.

$$\begin{pmatrix} x \\ y \\ 1 \end{pmatrix} \sim \begin{pmatrix} f_x & 0 & c_x \\ 0 & f_y & c_y \\ 0 & 0 & 1 \end{pmatrix} \begin{pmatrix} r_{11} & r_{12} & r_{13} & t_1 \\ r_{21} & r_{22} & r_{23} & t_2 \\ r_{31} & r_{32} & r_{33} & t_3 \end{pmatrix} \begin{pmatrix} X \\ Y \\ Z \\ 1 \end{pmatrix} \quad (1)$$

Considering the presence of moving objects and calibration errors, there may be some offset for each projection. To reduce erroneous labels, we run DBSCAN clustering [10] on each label projection to check whether the 3D points projected belong to a single cluster. If they do not, we only label the cluster with the most points.

### 4.5 Released data

We release the raw ROS Bagfiles, and extracted images, point cloud files, COLMAP [22] poses and sparse reconstructions. The raw data consists of a set of sequences, each of which is collected during a specific data recording session. To make the data more manageable, we divide each session into different subsequences or *"sectors"*, with each sector consisting of 1250 images and roughly 167 point cloud scans. In addition, we conducted face detection and applied blurring techniques to ensure privacy protection on campus.

**Multi-view images** Each image is named according to the convention `cam[id]-[timestamp].jpg`. We estimate synchronized timestamps for all images within a sector, using the method mentioned in section 3.3. The wide field of view (FoV) of 90 degrees for each of the five cameras results in

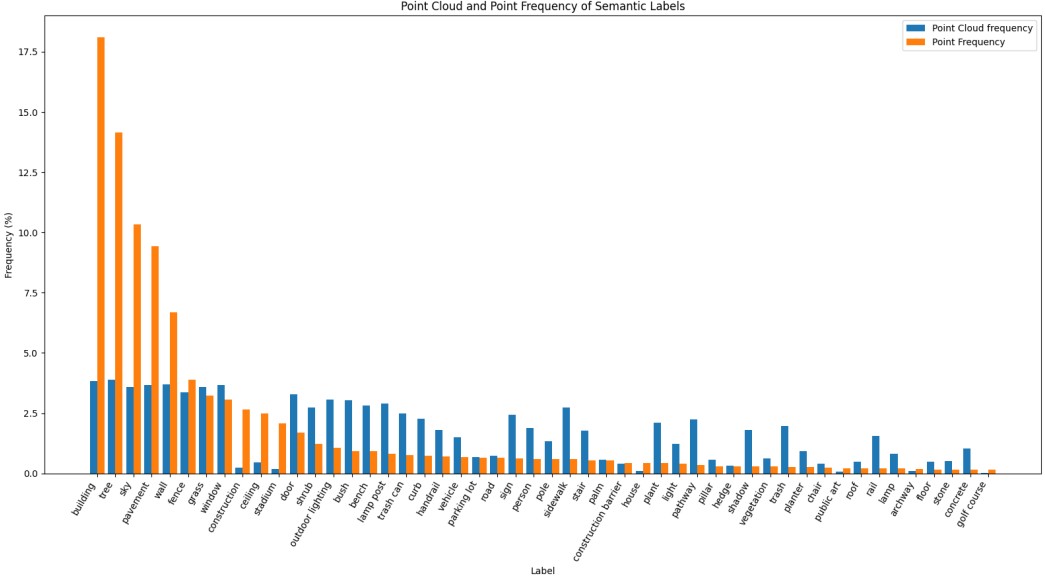

Figure 6: **Histogram of the semantic labels frequency in point cloud scans and points.** Top 50 frequently estimated semantic classes in points(orange), and correspoing point cloud scan frequency

significant overlap between their respective images, as depicted in Figure 1. This substantial overlap ensures more robust Structure from Motion (SfM) reconstruction. By having multiple views of the same scene, the SfM algorithm can triangulate feature points more accurately, leading to a more precise reconstruction of the 3D environment. This overlap also aids in improving the accuracy of semantic labelling. By leveraging overlapping information from multiple viewpoints, inconsistencies or errors in semantic annotations of 3D points from 2D-pixel maps can be identified and rectified through cross-validation. This double-checking mechanism helps to enhance the reliability of semantic labels assigned to objects in the scene.

**Semantic instances and masks for images** In addition to the raw image data, we also provide semantic labels and label masks generated by Grounded-SAM for each image in the dataset. These labels offer valuable insights into the semantic understanding of the scene, allowing researchers to perform tasks such as semantic segmentation and object detection.

**Semantically annotated 3D point cloud streams** As mentioned before, the pointcloud streams are captured at 10Hz. Similar to KITTI Semantic [4], we extract each of the pointcloud scans and annotate the 3D points by assigning semantic labels to individual points based on the closest image's label, using the method outlined in section 4.3. The color and corresponding label for each point are saved in a JSON file named `labels.json`, ensuring easy access and interpretation of the semantic annotations.

**Semantically annotated point clouds** In addition to the individual semantic annotated point cloud scans, we have processed each session's point cloud data using LeGO-LOAM [25] to generate a merged point cloud for each sector (area corresponding to the segments of a trajectory). We mention the statistics of the distribution of points in each of the point cloud scans and the merged point clouds in the supplemental material. Unlike the point cloud scans, sector-based point clouds have more points and offer a comprehensive overview of the semantic annotated scene. Through these semantic point clouds, researchers can gain deeper insights into the semantic structure and composition of the environment.

**Pose annotations for images.** We release interpolated poses from LeGO-LOAM, and COLMAP Structure from Motion (SfM) [22]. The COLMAP SfM results can serve as inputs for some generative model like NeRF or 3D Gaussian Splatting. Further, by utilizing the poses computed by COLMAP, we aim to improve the precision of our annotations given the different sampling rates of the LiDAR (10Hz) and cameras (15Hz). This alignment is crucial for accurately projecting semantic labels onto the 3D points based on the information extracted from the images. We are currently investigating

how to best merge the LiDAR and COLMAP poses, likely resulting in a unified set of poses indexed non-uniformly in time, for each image and for each point cloud. We expect that these unified poses will be released with the next version our dataset.

**Robotic dataset for visual navigation.** Our dataset comprises diverse sequences captured within a university environment, reflecting a range of real-world scenarios. Leveraging the compact form factor of our robot, we collected data across a variety of settings including roads, outdoor lobbies, ramps, and other typical campus landscapes. This dataset is particularly valuable for applications in visual navigation and is integrated into the comprehensive Open X-Embodiment dataset [7].

## 5 Benchmarks

### 5.1 Evaluation on Novel View Synthesis

We examine the current state-of-the-art (SOTA) Novel View Synthesis methods on several datasets: USCILab3D, ETH3D [23], Mip-NeRF360 [3], Tanks&Temples [12], and Deep Blending [12]. For each dataset, we run 3D Gaussian Splatting and evaluate the generated image quality using PSNR, SSIM, and L-PIPS metrics. For each scene, we use 7/8 of the data as the training set and 1/8 as the test set, then calculate the average result for each scene. Considering the large size of our dataset, we randomly extract one sector from each session to compute the average result.

Our dataset achieves superior PSNR, SSIM, and the best L-PIPS performance compared to other datasets (Table 5). Among these datasets, ours is the only one that provides large-scale scenes, making it suitable for a wider range of applications, such as simulators [2].

|                   | PSNR ↑ | SSIM ↑ | LPIPS ↓ | Resolution ↓ | interation |
|-------------------|--------|--------|---------|--------------|------------|
| USCILab3D (ours)  | 26.02  | 0.86   | 0.20    | 1280 × 720   | 7000       |
| ETH3D[23]         | 21.25  | 0.83   | 0.27    | 6048 × 4032  | 7000       |
| Tanks&Temples [15]| 21.20  | 0.77   | 0.28    | 980 x 540    | 7000       |
| Mip-NeRF360[3]    | 25.19  | 0.75   | 0.25    | 1256 x 828   | 7000       |
| Deep Blending[12] | 27.01  | 0.87   | 0.32    | 1332 x 876   | 7000       |

Table 5: **Performance comparison of 3D Gaussian splatting on different datasets.** Our dataset achieves superior performance compared to other datasets. Although Deep Blending demonstrates a higher PSNR, it only contains 2.6K images.

### 5.2 Evaluation on Semantic Segmentation and Completion

We also evaluate our dataset using key tasks: semantic segmentation, panoptic segmentation, and semantic scene completion. Semantic segmentation is crucial for understanding and labeling every point in a 3D point cloud with a specific class, providing detailed insights into the composition of the scene. Panoptic segmentation extends this by not only classifying each point but also distinguishing between different instances of the same class. This is particularly valuable for environments with multiple similar objects, enhancing the dataset's utility in more complex and dynamic scenarios. Lastly, semantic scene completion involves predicting the complete geometry and semantics of a scene, including occluded and unobserved regions. This task is vital for creating comprehensive and accurate representations of environments, which is indispensable for advanced applications in augmented reality and spatial analysis. We have included the results in the supplemental material.

## 6 Caveats

Thus far, our annotations have been machine-generated using the latest foundation models. Although this may pose a few risks, nevertheless, to the best of our knowledge, our method is the first of its kind to annotate 3D point clouds using image and text based foundational models without any manual intervention. Casual inspection by authors suggests that the annotations are indeed of high quality. However, we plan to validate them by hiring a group of human annotators to inspect and possibly

correct a fraction of the machine-generated annotations. We expects that this will be completed by the time of publication.

# 7 Discussion and Conclusion

In this paper, we introduced the USCILab3D dataset, a comprehensive outdoor 3D dataset designed to address the limitations of existing datasets in the domain of 3D scene understanding and navigation. Our dataset offers a diverse array of complex intersections and outdoor scenes meticulously collected across the USC University Park campus. With approximately 10 million images and 1.5 million dense point cloud scans, our dataset prioritizes intricate areas, enabling more precise 3D labelling and facilitating a broader spectrum of 3D vision tasks.

Moving forward, we believe that the USCILab3D dataset will serve as a valuable resource for researchers and practitioners across various domains, including computer vision, robotics, and machine learning. We anticipate that the dataset will stimulate further advancements in 3D vision-based models and foster the development of robust algorithms capable of tackling real-world challenges in outdoor environments.

# 8 Acknowledgments

This work was supported by the National Science Foundation (award 2318101), C-BRIC (one of six centers in JUMP, a Semiconductor Research Corporation (SRC) program sponsored by DARPA) and the Army Research Office (W911NF2020053). The authors affirm that the views expressed herein are solely their own, and do not represent the views of the United States government or any agency thereof.

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
