# Supplementary Material

## 1 Additional Implementation Details

### 1.1 Camera and LiDAR Calibration

We printed a checkerboard with a 9x10 grid of blocks, each measuring 87 mm x 87 mm. The calibration distance ranged from 1.3 m to 3 m. MATLAB software was used to run the calibration algorithm.

## 2 Additional Experiment details

For all the experiments for benchmark, we used a Core-10 desktop with 64-96 GB of memory and 1 3090-Ti GPU.

### 2.1 Model architectures and Hyperparameters

| Parameter | Value |
|-----------|-------|
| Model | Grounded-SAM |
| grounded_checkpoint | groundingdino_swint_ogc.pth |
| sam_checkpoint | sam_vit_h_4b8939.pth |
| box_threshold | 0.18 |
| text_threshold | 0.15 |

Table 1: Parameters for the Grounded-SAM model

| Parameter | Value |
|-----------|-------|
| Model Architecture | cylinder_asym |
| Output Shape | $256 \times 256 \times 32$ |
| Output Feature Dimension (out_fea_dim) | 256 |
| Number of Classes (num_class) | 6 |
| Use Normalization (use_norm) | True |
| Initialization Size (init_size) | 32 |
| Learning Rate | 0.001 |

Table 2: Parameters for the Semantic Segmentation model

### 2.2 Data for benchmark

To construct the train and test dataset for the above experiments, we randomly selected the following dates for benchmarking: 2023_07_05, 2023_07_11, 2023_08_08. The train dataset comprised of the data from the first 2 dates and the test dataset comprised of the data from the last date.

Submitted to the 38th Conference on Neural Information Processing Systems (NeurIPS 2024) Track on Datasets and Benchmarks. Do not distribute.

| Parameter | Value |
|-----------|-------|
| Model Architecture | Panoptic-PolarNet |
| Test Batch Size | 2 |
| Val Batch Size | 2 |
| Test Batch size | 1 |
| post proc threshold | 0.1 |
| post proc nms kernel | 5 |
| post proc top k | 100 |
| center loss | MSE |
| offset loss | L1 |
| center loss weight | 100 |
| offset loss weight | 10 |
| enable SAP | True |
| SAP start epoch | 30 |
| SAP rate | 0.01 |

Table 3: Parameters for Panoptic Segmentation model

| Parameter | Value(s) |
|-----------|----------|
| Model Architecture | 4D-StOP |
| Learning Rate | 0.0005 |
| Momentum | 0.98 |
| Stride | 1 |
| Max in points | 5000 |
| Sampling | importance |
| Decay Sampling | None |
| Input Threads | 16 |
| Checkpoint Gap | 100 |

Table 4: Parameters for the 4D Panoptic Segmentation model

# 3 Baselines

We use mean intersection-over-union (mIoU) percentages and intersection-over-union (IoU) percentages provided by SemanticKITTI website as the baseline to compare the models' performances on the SemanticKITTI dataset and our dataset. Table 6 presents the mIoU percentages on various tasks, each with a model we would use in our experiments. The data is provided by the SemanticKITTI website.

# 4 Benchmark

We divide the 267 labels to 6 and 11 categories and produce benchmark scores on these two sets of categories.

## 4.1 Semantic Segmentation

**Tasks** In semantic segmentation of point clouds, we want to infer the label of each three-dimensional point. Therefore, the input to all evaluated methods is a list of coordinates of the three-dimensional points along with their remission, i.e., the strength of the reflected laser beam which depends on the properties of the surface that was hit. Each method should then output a label for each point of a scan, i.e., one full turn of the rotating LiDAR sensor.

| Parameter | Value(s) |
|-----------|----------|
| Model Architecture | MF-MOS |
| Learning Rate | 0.002 |
| Learning Rate Decay | 0.99 |
| Momentum | 0.9 |
| EpsilonW | 0.001 |
| Number of Input Scans | 8 |

Table 5: Parameters for the Moving Object Segmentation model

| Task | Model | mIoU (%) |
|------|-------|----------|
| Semantic Segmentation | Cylinder3D | 67.8 |
| Panoptic Segmentation | Panoptic-PolarNet | 59.5 |
| 4D Panoptic Segmentation | 4D-StOP | 58.8 |

Table 6: Models of various tasks used in our experiments and their performances on SemanticKITTI

**Metrics** To assess the labeling performance, we used mean Jaccard Index or mean intersection-over-union (mIoU) metric over all classes, given by

$$mIoU = \frac{1}{C} \sum_{c=1}^{C} \frac{TP_c}{TP_c + FP_c + FN_c},$$ (1)

where $TP_c$, $FP_c$, and $FN_c$ correspond to the number of true positive, false positive, and false negative predictions for class $c$, and $C$ is the number of classes.

**Method** The segmentation is performed using Cylinder3D with batch size for training is 2, and the batch size for test is 1, trained over 200 epoches..

**Result** Table 7 presents the mean intersection-over-union (mIoU) percentages for various categories in our dataset. The results reveal a significant variance in performance across different categories. Notably, 'Structure' and 'Ground' both achieved high mIoU at $89.10\%$ and $90.12\%$, 'Nature' show slightly lower mIoU with value $85.03\%$. The rest are 'Vehicle', 'General Objects' and 'Sidewalk Objects' with values of $72.06\%$, $57.66\%$ and $54.16\%$, respectively, and the model is still able to distinguish the categories with relative high mIoU. The overall average mIoU is $74.69\%$, which points to a significant gap in achieving high accuracy across all categories.

| Category | mIoU (%) |
|----------|----------|
| Vehicle | 72.06 |
| Nature | 85.03 |
| Ground | 90.12 |
| Structure | 89.10 |
| Sidewalk Objects | 54.16 |
| General Objects | 57.66 |
| Average | 74.69 |

Table 7: Mean Intersection over Union (mIoU) percentages of 6 major categories for semantic segmentation task.

## 4.2 Panoptic Segmentation

**Tasks** In panoptic segmentation of point clouds, we want to infer the label of each three-dimensional point and the instance of so-called thing classes. Therefore, the input to all evaluated methods is a list of coordinates of the three-dimensional points along with their remission, i.e., the strength of the reflected laser beam which depends on the properties of the surface that was hit. Each method should then output a label for each point of a scan, i.e., one full turn of the rotating LiDAR sensor.

**Metrics** We use the panoptic quality (PQ) proposed by Kirillov et al. defined by

$$\frac{1}{C} \sum_{c=1}^{C} \frac{\sum_{(S,\hat{S}) \in TP_c} IoU(S, \hat{S})}{|TP_c| + \frac{1}{2}|FP_c| + \frac{1}{2}|FN_c|} \tag{2}$$

where $TP_c$, $FP_c$, and $FN_c$ correspond to the number of true positive, false positive, and false negative predictions for class $c$, and $C$ is the number of classes. A match between segments is a true positive if their IoU (intersection-over-union) is larger than 0.5. To account for segments of stuff classes that have multiple connected components, Porzi et al. proposed a modified metric PQ† that uses just the IoU for stuff classes without distinguishing between different segments.

**Method** The completion is performed using Panoptic-PolarNet with batch size for training is 2, and the batch size for test is 2, trained over 50 epoches.

**Result** The results are shown in Table 8. presents the mean intersection-over-union (mIoU) percentages for various categories in our dataset. The results reveal a significant variance in performance across different categories. Notably, 'Structure' achieved the highest mIoU at 60.37%, 'Nature', 'Ground', 'Sidewalk Objects' and 'Vehicle' show slightly lower mIoU values of 21.56%, 18.81%, 15.96% and 14.70%, respectively. 'General Objects' category have the lowest mIoU, 0.88%, highlighting the difficulty in segmenting these less defined and diverse classes. The overall average mIoU is 22.046%. The ranking of the performance of each categories behave very similar to semantic segmentation, the reason is that the dataset contains a large portion of data that belongs to construction, while the other categories such as 'vehicle' and 'general objects' consists of a smaller portion of the dataset.

| Category | mIoU (%) |
|---|---|
| Vehicle | 14.70 |
| Nature | 21.56 |
| Ground | 18.81 |
| Structure | 60.37 |
| Sidewalk Objects | 15.96 |
| General Objects | 0.88 |
| Average | 22.046 |

Table 8: Mean Intersection over Union (mIoU) percentages of 6 major categories for panoptic segmentation task.

## 4.3 4D Panoptic Segmentation

**Task** The task of 4D-panoptic segmentation is to assign a unique instance ID in addition to inferring the semantic label for each three-dimensional point in a sequence of scans (a scan is a full rotation of the LiDAR sensor). This allows instance segmentation and object tracking to be combined with semantic segmentation into a single task. The inputs of this task are coordinates of 3D-points and the remission of the corresponding points. The remission is the strength of the reflected laser beams, which depends on the surface they were reflected from. The output of the task should be, for each point, a semantic label and instance ID.

**Method** We perform experiments of this task using 4D-StOP, a panoptic segmentation model for 4D LiDAR. The experiemts are conducted with batch size 8 for training, and batch size 1 for validation, pretrained over 800 epochs and trained over 300 epochs. While training for 300 epochs, the semantic segmentation parameters are frozen to learn high-quality geometric features. We conducted two experiments; in each experiment the dataset is divided into 17 and 6 categories, respectively, while all other hyperparameters remain the same.

**Metrics** To assess the labeling performance, we used intersection-over-union (IoU) metric over all classes, given by

$$IoU = \frac{TP_c}{TP_c + FP_c + FN_c}, \tag{3}$$

**Result** Tables 9 and 10 present the intersection-over-union (IoU) percentages for various categories in our dataset. The dataset is divided into 17 and 6 categories, respectively. Among the categories, those related to structures and nature stands out with the highest IoUs across both experiments, indicating robust segmentation accuracy in identifying architectural elements, buildings, trees, and grass. Conversely, the 'Vehicle' category exhibit lower IoU values across both experiments, suggesting challenges in accurately segmenting vehicles. In some categories, such as 'Ground', the model performs better if the category is divided into more specific groups, such as 'Grass and Natural Ground' and 'Roads', as opposed to grouping anything related to ground as a single category.

| Category | IoU (%) |
|---|---|
| Light | 0.00 |
| Barriers | 15.53 |
| Buildings and Structures | 59.53 |
| Statues | 0.07 |
| Objects | 5.33 |
| Furniture | 4.20 |
| Environment | 0.42 |
| Plants | 48.56 |
| Grass and Natural Ground | 40.89 |
| People | 0.81 |
| Vehicle | 0.00 |
| Roads | 45.67 |
| Road Signs | 0.00 |
| Drainage Covers | 0.00 |
| Sidewalks | 0.09 |
| Shadow | 0.00 |
| Water | 13.82 |
| Average | 38.01 |

Table 9: Intersection over Union (IoU) percentages for 17 categories on 4D Panoptic Segmentation.

| Category | IoU (%) |
|---|---|
| Vehicle | 0.00 |
| Nature | 49.07 |
| Ground | 2.55 |
| Structure | 74.62 |
| Sidewalk Objects | 73.80 |
| General Objects | 4.95 |
| Average | 34.17 |

Table 10: Intersection over Union (IoU) percentages for 6 categories on 4D Panoptic Segmentation.

## 4.4 Moving Object Segmentation

**Task** The task of moving object segmentation is to assign a motion label for each three-dimensional point in a scan (a full rotation of the LiDAR sensor). The inputs of this task are coordinates of 3D-points and the remission of the corresponding points. The remission is the strength of the reflected laser beams, which depends on the surface they were reflected from. The output of the task should be a motion label for each point in the scan. In this experiment, we set up the model to distinguish movable objects (for example, vehicles) from immovable ones (for example, structures). Due to limitations we did not conduct experiments on distinguishing moving objects.

**Method** The experiment is performed using the MF-MOS model with batch size 4 for training, and the model is trained for 150 epochs.

**Metrics** To assess the labeling performance, we used intersection-over-union (IoU) metric over all classes, given by

$$IoU = \frac{TP_c}{TP_c + FP_c + FN_c},$$ (4)

**Result** Table 11 presents the intersection-over-union (IoU) percentages for immovable and movable categories. The IoU is high for immovable objects but very low for movable objects, suggesting that the model has trouble with identifying movable objects when the objects are not actually moving.

| Category | IoU (%) |
|----------|---------|
| Immovable | 84.75 |
| Movable | 2.49 |
| Average | 43.62 |

Table 11: Intersection over Union (mIoU) percentages on Moving Object Segmentation.

Overall, the performance across these tasks underscores the challenges posed by our dataset's complexity, with 267 label categories condensed into 6 predicted categories. The categorization decision may have affected the model's ability to distinguish finer details within each category. With our dataset, future work can focus on improving the model's capacity to handle such diverse and complex categories, potentially by incorporating more sophisticated network architectures or additional data augmentation techniques. Besides that, although all categories in the dataset consists of many data points, but the ratio between different categories can have significant difference, for example, the data points of building and tree are the two most frequency classes in the dataset, this explain why the mIoU of "Structure" and "Nature" are higher than the others. The future work will include using the resampling techniques and class weighting to overcome the imbalance issue in the dataset.

# 5 Additional Dataset details

## 5.1 Dataset Source

The raw data, processed data, and framework code can be found on our website.

## 5.2 Motivation

The dataset was created to enable research on 3D computer vision tasks, including large-scale 3D reconstruction, and semantic point clouds tasks. Additionally, we developed a pipeline for automatic semantic labeling, which is essential for unsupervised large-scale data training.

The dataset pipeline was created by Kiran Lekkala and Henghui Bao at University of Southern California.

## 5.3 Composition

### 5.3.1 Metadata

The metadata consists of bag files, with each bag file corresponding to a session from one camera. Each camera's bag file contains the Velodyne LiDAR information. The file All_Sessions.txt records the date of each session and the names of the five bag files.

### 5.3.2 Processed data

The format of processed data is outlined on the website.

### 5.4 Maintenance

The dataset will be available for download from our server and Google Drive. It will be continuously updated with more accurate labels and additional data. For any inquiries, please contact klekkala@usc.edu. If you wish to contribute to the dataset, please reach out to the original authors.

### 5.5 Distribution

The dataset was released in 2024 without a DOI and publicly available on the internet and distributed on our website.

### 5.6 License

Our dataset follows the CC BY 4.0 license.