# OpenReview forum: "USCILab3D: A Large-scale, Long-term, Semantically Annotated Outdoor Dataset"
_NeurIPS.cc/2024/Datasets_and_Benchmarks_Track — NeurIPS 2024 Track Datasets and Benchmarks Poster_

### Official Review · Reviewer_2uvC · 2024-07-15
**Comments**

**Rating:** 6
**Confidence:** 3
**Correctness:** The claims made are correct.
**Clarity:** The paper is well written.

**Review:**

This project contributes to the development of a large-scale outdoor dataset featuring multi-view images and LiDAR information. The authors utilize large foundation models for annotation, enhancing the dataset with detailed information about the scenes. This comprehensive approach not only includes diverse perspectives captured through multiple views but also integrates advanced LiDAR data to offer precise spatial details. By leveraging state-of-the-art foundation models, the authors ensure that the annotations are rich and informative, providing deeper insights into the scenes and facilitating a wide range of applications in research and development.

**Strengths:**

The paper presents the creation of a large-scale 3D outdoor dataset comprising 10 million images and 1.4 million LiDAR scans. This dataset surpasses previous datasets in terms of size and comprehensiveness. It includes not only images and LiDAR point clouds but also rich semantic information.

The authors employ foundation models such as SAM and GPT-4 for data annotation, demonstrating a scalable approach to generating detailed and accurate annotations. These models enhance the dataset by providing deeper semantic insights, making it more valuable for various applications in research and development.

The paper clearly explains the dataset collection process, detailing the methodologies and technologies used to gather and integrate the multi-modal data. Additionally, the authors conduct novel view synthesis experiments to evaluate the performance of different datasets, providing valuable benchmarks and insights into the effectiveness of the proposed dataset.

**Additional Feedback:**

See comments above.

**Documentation:**

The details are sufficient.

**Ethics:**

There is no ethical concerns.

**Limitations:**

Yes, the authors adequately addressed the limitations and potential negative societal impact of their work.

**Opportunities For Improvement:**

Further evaluation of the proposed dataset is necessary. This includes exploring additional novel view synthesis methods beyond 3D Gaussian Splatting to fully assess the dataset's potential.

The authors should also provide more details on how they compute the incorrect pixel labels presented in Figure 5, as this is crucial for understanding the dataset's accuracy and reliability.

To enhance the paper, it would be beneficial to discuss other tasks for which the dataset can be used. Potential applications include object detection, scene segmentation, autonomous vehicle navigation, and urban planning. By outlining these additional use cases, the authors can demonstrate the versatility and broad applicability of their dataset, further underscoring its value to the research community.

The experimental design of Table 3 is unconventional. Typically, different methods are compared on the same dataset to evaluate their performance. However, in this case, the performance of a single method is being compared across different datasets. This approach makes it challenging to assess the quality of the datasets, as the quality cannot be adequately determined using a single method. It would be more informative to compare multiple methods on each dataset to provide a clearer evaluation of both the methods and the datasets.

**Relation To Prior Work:**

The paper clearly discussed the difference from previous contributions.

**Summary And Contributions:**

In this paper, the authors present a large-scale, semantically annotated outdoor dataset. The dataset was collected using a fleet of four mobile robots, each equipped with five cameras and a 32-beam, 360° scanning LIDAR. The data is meticulously annotated using state-of-the-art large foundation models. This comprehensive dataset includes 1.4 million point clouds and 10 million images. Additionally, the authors evaluate a novel view synthesis method on this dataset, providing valuable insights into its applicability and performance in generating new perspectives from the acquired data.

---

> ### Author Rebuttal · Authors · 2024-08-17
>
> We want to thank reviewer 2uvC for taking the time to provide such constructive feedback.
>
> Here is our rebuttal to **Opportunities for improvement**.
>
> 1. Thank you for the suggestion. We have experimented with other novel view synthesis methods, such as F2-NeRF and Instant-360. However, due to their lower PSNR performance on most datasets, we initially focused on presenting the 3D Gaussian Splatting results. We will include the results of these additional methods in future updates to provide a more comprehensive evaluation.
>
> 2. Figure 5 illustrates the impact of unrelated prompts on Grounded-SAM by comparing its output when provided with additional unrelated prompts versus its output when given the exact prompt. Regarding the dataset’s accuracy and reliability, we are constantly optimizing and refining our annotation strategy. We conducted a small-scale verification on projections from 500 scans, achieving an accuracy rate of **85.11% (2264/2660)** correct object labels. And we are currently using Amazon Mechanical Turk for a detailed analysis and iterative correction. We will soon release the results of this manual analysis, and the human-corrected labels, to provide greater transparency and reliability in our dataset.
>
> 3. We appreciate your suggestion to discuss additional applications of our dataset. We will highlight its potential use in tasks such as 3D object detection, 3D scene segmentation, autonomous vehicle navigation, and urban planning. For example, the dataset could be particularly valuable for 3D segmentation models like Point-SAM, which operate on similar principles as SAM but have not achieved comparable results, partly due to the limited scale of available datasets. With a larger dataset, we believe that models can achieve better generalization. For visual navigation, as stated in the Open-X/RT-X paper, we could train a large-scale Robot Transformer (RT-X) model using the offline dataset of visual navigation data. Essentially, this data consists of a stream of observation, along with an action label, that is obtained from human teleoperation. For more details, please check out the RT-X paper (https://arxiv.org/pdf/2310.08864)
>
> 4. Thank you for your suggestion. Yes, we wanted to show that, our dataset is a better benchmark, than the others, assuming that we are comparing the performance of a single 3D rendering method on all the datasets. But you have made a valid point, and currently, we are in the process of training two more methods for each dataset and will update the results in Table 3 soon.

---

### Official Review · Reviewer_12vN · 2024-07-22
**Review of USCILab3D**

**Rating:** 6
**Confidence:** 3
**Clarity:** This paper is well written.

**Review:**

Please refer to the `Strengths` and `Opportunities For Improvement` for pros and cons.

**Strengths:**

1. This paper proposes a realistic outdoor dataset containing multi-view video sequences and LiDAR points, which is valuable for novel view synthesis and point cloud segmentation. The huge number of multi-view video frames are also valuable for generative models.

2. It is a good practice to annotate semantic labels with the help of foundation models. The sufficient semantic labels could be valuable for many downstream tasks.

3. This paper is clear to read and easy to follow.

4. The authors conduct experiments on two applications on the proposed dataset, which demonstrate the use cases of USCILab3D dataset.

**Additional Feedback:**

My main concerns lie in the motivation of this work. I wish to get reponse of the following two questions:

1. Given the outdoor datasets like Waymo, nuScenes, SemanticSTF and WildScenes, what's the necessity to introduce this dataset?
2. Why do we need the fine-grained semantic annotations proposed in this work?


After the author rebuttal and discussion phase, I'd like to raise my score from 5 to 6.

**Correctness:**

The dataset is constructed in a sound way. But the novel view synthesis experiments could not verify the superiorness of this dataset.

**Documentation:**

The details of data collection, a URL for reviewer to access the dataset, a hosting website, and a maintenance plan are provided.

On the other hand
1) The organization of this dataset is unclear.
2) The liscense is missing.
3) The ethical and responsible use claim is missing.

**Ethics:**

This dataset could contain human subjects since the robot collects the data in outdoor scenes. And the authors conducted face detection and applied blurring techniques to ensure privacy protection on campus.

**Limitations:**

The direct utilization of foundation model outputs could limit the accuracy of semantic annotations.

**Opportunities For Improvement:**

1. The related works mentioned by this paper omits a series of work which include semantic point cloud annotations for urban scenes, such as nuScenes [1], waymo [2] and SemanticSTF [3]. There are also datasets that contain semantic point cloud ground truths in general outdoor scenes like WildScenes [4]. A comprehensive comparison to these datasets could help readers to understand the novelty and importance of the proposed  USCILab3d dataset.

2. One possible advantage of the proposed USCILab3d dataset is the sufficient and fine-grained semantic labels. For this point, I suggest the authors to discuss and verify the necessity of fine-grained semantic labels. For example, the authors should answer these questions: 1) why do we need so many fine-grained semantic labels? 2) Could these fine-grained annotations enable new applications or new features?

3. A direct comparison on the semantic classes would be necessary if the fine-grained semantic labels are a main contribution, i.e., the authors should consider comparing the semantic classes in different datasets.

4. The Grounded-SAM outputs are not always accurate. It would be better if the authors design some strategies to correct and improve the semantic annotations.

5. In the novel view synthesis experiments, the authors claim that `Our dataset achieves superior performance compared to other datasets.` How could this experiment demonstrates the dataset superiority? And what kind of superiority could this experiment show?

6. Minor:
1) What's the differences between `point cloud frequency` and `point frequency` In Figure 6?
2) Table 3, interation >> iteration

[1] Fong, W. K., Mohan, R., Hurtado, J. V., Zhou, L., Caesar, H., Beijbom, O., & Valada, A. (2022). Panoptic nuscenes: A large-scale benchmark for lidar panoptic segmentation and tracking. IEEE Robotics and Automation Letters, 7(2), 3795-3802.

[2] Sun, P., Kretzschmar, H., Dotiwalla, X., Chouard, A., Patnaik, V., Tsui, P., ... & Anguelov, D. (2020). Scalability in perception for autonomous driving: Waymo open dataset. In Proceedings of the IEEE/CVF conference on computer vision and pattern recognition (pp. 2446-2454).

[3] Xiao A, Huang J, Xuan W, et al. 3d semantic segmentation in the wild: Learning generalized models for adverse-condition point clouds[C]//Proceedings of the IEEE/CVF Conference on Computer Vision and Pattern Recognition. 2023: 9382-9392.

[4] Vidanapathirana, K., Knights, J., Hausler, S., Cox, M., Ramezani, M., Jooste, J., ... & Moghadam, P. (2023). WildScenes: A Benchmark for 2D and 3D Semantic Segmentation in Large-scale Natural Environments. arXiv preprint arXiv:2312.15364.

**Relation To Prior Work:**

Comparisons to nuScenes [1], waymo [2], SemanticSTF [3], and WildScenes [4] are missing.

[1] Fong, W. K., Mohan, R., Hurtado, J. V., Zhou, L., Caesar, H., Beijbom, O., & Valada, A. (2022). Panoptic nuscenes: A large-scale benchmark for lidar panoptic segmentation and tracking. IEEE Robotics and Automation Letters, 7(2), 3795-3802.
[2] Sun, P., Kretzschmar, H., Dotiwalla, X., Chouard, A., Patnaik, V., Tsui, P., ... & Anguelov, D. (2020). Scalability in perception for autonomous driving: Waymo open dataset. In Proceedings of the IEEE/CVF conference on computer vision and pattern recognition (pp. 2446-2454).
[3] Xiao A, Huang J, Xuan W, et al. 3d semantic segmentation in the wild: Learning generalized models for adverse-condition point clouds[C]//Proceedings of the IEEE/CVF Conference on Computer Vision and Pattern Recognition. 2023: 9382-9392.
[4] Vidanapathirana, K., Knights, J., Hausler, S., Cox, M., Ramezani, M., Jooste, J., ... & Moghadam, P. (2023). WildScenes: A Benchmark for 2D and 3D Semantic Segmentation in Large-scale Natural Environments. arXiv preprint arXiv:2312.15364.

**Summary And Contributions:**

This paper introduces a realistic large-scale outdoor dataset. The dataset is collected by a robot mounted with 5 cameras and 1 LiDAR sensor, containing multi-view video sequences and annotated semantic LiDAR point clouds.

To annotate the semantic labels of collected point clouds, the authors utilize GPT4 to generate semantic text prompts and utilize Grounded SAM to segment contents according to text prompts. These foundation models help to save a lot of labeling cost.

Based on the USCILab3D dataset, the authors conducted experiments of novel view synthesis and point cloud segmentation.

---

> ### Author Rebuttal · Authors · 2024-08-17
>
> We want to thank reviewer 12vN for taking the time to provide such constructive feedback.
>
> **Opportunities for Improvement:**
>
> 1. Thank you for your suggestion. We will include a comparison with the mentioned datasets, such as nuScenes, Waymo, SemanticSTF, and WildScenes. This addition will better highlight the advantages and unique contributions of our USCILab3d dataset.
>
> | **Dataset name**  | **Point Clouds** | **Semantic Labels** | **Semantic classes**                                                                                                      |
> |-----------------------|---------------------------|------------------------------|------------------------------------------------------------------------------------------------------------------------------------|
> | Our Dataset (USCILab3D)  | 1.4M                      | 267                          | Vehicle, nature, human, ground, structure, street furniture, architectural elements, signs and symbols, general objects, lightning |
> | nuScenes    | 390K                      | 31                           | vehicle, human, animal, movable object, flat, static                                                                               |
> | Waymo motion | 230K                      | 23                           | Traffic Entities: Car, Truck, Bus, Motorcyclist, Bicyclist, Pedestrian, etc.                                                       |
> | SemanticSTF  | 2K                        | 21                           | flat, construction, nature, vehicle, human, object                                                                                 |
> | WildScenes   | 12K                       | 15                           | terrain, vegetation, object, structure, water, sky                                                                                 |
>
>
> 2. We believe that large-scale fine-grained semantic labels are crucial for advancing 3D vision tasks, such as semantic segmentation. For instance, in 2D vision, Segment Anything has significantly impacted various downstream tasks: object tracking, language segmentation, image editing. However, 3D segmentation models like Point-SAM, which follow similar logic, have not yet achieved comparable results. One contributing factor is the lack of large-scale datasets. Current models are often trained on smaller datasets like PartNet, ScanNet, and Fusion360. We believe that with a larger dataset, models can achieve better generalization.
>
> 3. We will add a comparison of semantic classes across different datasets to underscore the significance of the fine-grained semantic labels in our dataset. A brief comparison can be found in the table above.
>
> 4. We acknowledge that machine-generated annotations may contain inaccuracies. We conducted a small-scale verification on projections from 500 scans, achieving an accuracy rate of **85.1% (2264/2660)** correct object labels. To address this, we are optimizing our annotation strategies and employing Amazon Mechanical Turk for a detailed analysis and iterative correction.
>
> 5. Our dataset’s higher sampling frequency allows it to achieve a higher PSNR, resulting in highly realistic reconstructions. This realism is beneficial for downstream tasks. For example, in our experiments, we used the reconstructed scenes to build a simulator. The high PSNR enables seamless navigation within the reconstruction, including moving to areas not directly recorded.
>
> 6. Minor: **(a)** In Figure 6, “point cloud frequency” refers to how often a label appears across different scans, while “point frequency” refers to how frequently a label appears across all pixels. **(b)** Thanks for the feedback.
>
> **Additional Feedback:**
> 1. The USCILab3d dataset focuses on navigation environments typical of urban settings, such as sidewalks and pedestrian areas, rather than road driving. This emphasis improves understanding of non-driving environments. Compared to other datasets, ours is larger in scale, making manual annotation challenging. Our method aims to create a large-scale dataset that, while potentially sacrificing some accuracy, is invaluable for training large models, which can lead to better generalization.
>
> 2. As discussed in (Opportunities for Improvement, item 2), fine-grained annotations enable new applications and features, further advancing the field.
>
>
> Also, thanks for the missing citations. We added them in our local copy of the manuscript.

---

> > ### Comment · Reviewer_12vN · 2024-08-27
> >
> > I appreciate the authors' feedback which effectively addresses most of my concerns. I agree that a comprehensive 3D dataset focusing on sidewalks and pedestrian areas would be immensely beneficial to the community. The dataset's geographical coverage within the campus also ensures a good scene diversity, which is crucial for its utility.
> >
> > Regarding the dataset's purported superiority in novel view synthesis, I recommend that the authors consider incorporating point 5 from the rebuttal into Section 5.1 and enhance the clarity of the experimental result explanations. A more detailed elucidation of the dataset's advantages would be highly beneficial compared to simply stating 'superior performance'. The current rendition of Section 5.1 might be somewhat perplexing for researchers engaged in 3D reconstruction.
> >
> > I'd like to raise my score to 6: Marginally above the acceptance threshold.

---

> > > ### Author Response · Authors · 2024-08-27
> > > **Official comment from the authors**
> > >
> > > Thanks to reviewer 12vN for making such valuable comments. Yes, we are currently working on updating Section 5.1 by incorporating PSNR and higher sampling frequency (point number 5) along with a more detailed evaluation. We thank the reviewer for raising the score to 6. However, it seems like you need to update the original score and would deeply appreciate it if you could do so. Thank you!

---

### Official Review · Reviewer_RXE8 · 2024-07-24
**Outdoor dataset for the mobile-robot navigation**

**Rating:** 8
**Confidence:** 4
**Correctness:** Yes
**Clarity:** Yes

**Review:**

This paper is well written and easy to follow. The released dataset is well structured thus easy to utilize, and already is a part of Open X-Embodiment.

**Strengths:**

Most existing datasets are designed for autonomous driving and primarily focus on driving environments, making their annotations less suitable for mobile robot platforms in general navigation scenarios within smart cities. The proposed dataset offers navigation environments typical of urban settings, such as sidewalks and pedestrian areas, rather than road driving. This helps improve the understanding of non-driving environments. Additionally, it provides labeling using GPT models and semantic segmentation through the "segment anything" approach, making it highly relevant for visual-language navigation and the use of LLMs in robotic navigation.

**Additional Feedback:**

None

**Documentation:**

Yes

**Limitations:**

Discussed above

**Opportunities For Improvement:**

For robot navigation, it is essential to display the entire trajectory on a satellite image to understand the scale of the environment where the dataset was acquired. Given that this dataset is intended for navigation tasks, providing ground truth trajectories and estimated trajectories inferred from existing LiDAR localization methods is crucial. While labels from GPT models aid in semantic scene understanding, additional considerations for practical navigation are necessary, such as incorporating traversability information. It is also important to detail aspects of the mobile platform, including robot specifications, camera calibration matrix, IMU-to-LiDAR and camera-to-LiDAR transform matrices, to ensure comprehensive and effective use of the dataset for navigation purposes.

**Relation To Prior Work:**

Yes

**Summary And Contributions:**

This paper introduces a dataset which is a large-scale, annotated outdoor dataset collected over a year within the USC campus using a mobile robot equipped with cameras and a 360° LIDAR. This dataset includes 1.4 million point clouds and 10 million images, annotated with state-of-the-art models for multi-view imagery, 3D reconstructions, and semantic categories. The dataset focuses on intricate intersections and dense multi-view scenes, enabling precise 3D labeling and supporting diverse applications in computer vision, robotics, and machine learning. It aims to facilitate advancements in 3D vision tasks and the development of robust algorithms for real-world outdoor environments.

---

> ### Author Rebuttal · Authors · 2024-08-17
>
> We thank the reviewer RXE8 for providing such valuable feedback.
>
> Thank you for your very insightful comments. We are indeed considering the opportunities for improvement which you suggested, as guidance for our ongoing and future work.
>
> In regards to providing satellite images (bird's eye view) of the trajectories, we are currently working on using our algorithm involving Iterative closest point (ICP) [1] for estimating global poses of the images/pointcloud scans that could then be projected on a satellite image.
>
>
> [1] Least-square fitting of two 3-D point sets". IEEE Pattern Analysis and Machine Intelligence. 9 (5): 698–700

---

### Official Review · Reviewer_8QQu · 2024-07-24

**Rating:** 5
**Confidence:** 5

**Review:**

The USCILab3D dataset presents a contribution to the fields of computer vision and robotics. It stands out due to its scale, the variety of annotations, and the high quality of the data collected over diverse environmental conditions. The use of advanced techniques such as GPT-4 and Grounded-SAM for semantic annotation enhances the dataset's utility.

## Pros
Scale and Diversity: Compared to earlier outdoor datasets, USCILab3D distinguishes with a more extensive collection of frames and a wider variety of environmental conditions.
Quality of Annotations: High-quality semantic annotations across 267 categories.
Technological Integration: Utilizes advanced models like GPT-4 and Grounded-SAM for semantic labeling, ensuring robust and accurate annotations.

## Cons
Limited Geographical Scope: While comprehensive, the dataset is confined to the USC campus, which may limit its applicability to other environments.
Potential Annotation Errors: Despite employing sophisticated models, machine-generated annotations may contain inaccuracies. Have you conducted comparisons with human-labeled annotations throughout the process?

**Strengths:**

The dataset's primary strengths lie in its extensive coverage and high-quality annotations. Its detailed semantic labels and comprehensive multi-view imagery facilitate a wide range of research applications, from novel view synthesis to 3D scene understanding. The integration of state-of-the-art models for annotation ensures accuracy and relevance, making it a valuable resource for the broader research community. The ethical considerations, such as privacy protection through face blurring, reflect a commitment to responsible data usage.

**Additional Feedback:**

None

**Clarity:**

The composition of this paper necessitates further refinement.

## Standardizing stylistic choices
Consistency in punctuation, such as the decision to place a period after titles—including figure titles (e.g., Fig. 2 & Fig. 3), table titles (e.g., Tab. 2 & Tab. 3), and chapter headings (as seen in Section 4.5)—should be maintained throughout the document.

## Visual presentation
notably by increasing the font size in figures for enhanced readability, as observed in Figures 2 and 5.

**Correctness:**

The claims made in the submission are well-supported by the data and methodologies presented.

**Documentation:**

The dataset is well-documented, with sufficient details on data collection, organization, and annotation processes. Ethical considerations, such as privacy protection, are addressed. The inclusion of documentation for the intended uses and hosting plans ensures the dataset's sustainability and responsible use.

**Limitations:**

The authors have acknowledged the limitations related to the geographical scope and potential annotation errors. They plan to address these by involving human annotators for verification, which is a commendable step.

**Opportunities For Improvement:**

Geographical Expansion: Extending the dataset to include other geographical areas would enhance its generalizability.
Manual Verification: Incorporating a phase of manual verification for semantic annotations could improve accuracy and reliability.
Detailed Error Analysis: Providing a detailed error analysis of the machine-generated annotations would help users understand potential limitations and improve their models accordingly.

**Relation To Prior Work:**

The paper clearly discusses how the USCILab3D dataset differs from and improves upon existing datasets. It provides a comprehensive comparison with other multi-view and semantic scene datasets.

**Summary And Contributions:**

USCILab3D dataset contributes a large-scale, long-term, semantically annotated outdoor dataset. It includes 10 million images and 1.4 million point clouds collected over a year on the USC campus using a mobile robot with multiple cameras and a 32-beam LiDAR. The dataset offers comprehensive multi-view imagery, 3D reconstructions, and semantically annotated images and point clouds across 267 categories. The dataset is also benchmarked on novel view synthesis and semantic segmentation tasks. It addresses gaps in existing 3D datasets by providing high-quality, dense annotations and diverse environmental conditions.

---

> ### Author Rebuttal · Authors · 2024-08-17
>
> We want to thank the reviewer 8QQu for taking the time to provide such constructive feedback.
>
> 1. We acknowledge that machine-generated annotations may contain inaccuracies, and we are actively working to mitigate this issue. We conducted a small-scale verification on projections from 500 scans, achieving an accuracy rate of **85.11% (2264/2660)** correct object labels. While this accuracy may not be perfect, given the dataset's large scale, it remains highly valuable for training large models despite some inherent errors.
>
> 2. We are now in the process of utilizing Amazon Mechanical Turk to identify and analyze mislabeled data. This detailed analysis will help us enhance the accuracy of our annotations.
>
> 3. We will soon release an updated version of the dataset that incorporates human-verified annotations to improve its reliability.
>
> 4. About Geographical Expansion, thank you for your suggestion. We are still in the process of collecting more data and will soon update the dataset.
>
> 5. Thanks for the suggestions regarding standardizing stylistic choices and visual presentation. We've incorporated them in our local copy of the manuscript.

---

> > ### Comment · Reviewer_8QQu · 2024-08-18
> > **point 1 ground**
> >
> > it remains highly valuable for training large models despite some inherent errors.
> >
> > I think this point should be grounded with experimental results.
> >
> > Another paper in the batch (whose submission number is 981) shows that pre-training on the proposed dataset (also automatically labeled and checked by human annotators) can improve semantic understanding on ScanNet over latest state-of-the-art methods. I think with this kind of results, one can claim 'it remains highly valuable for training large models despite some inherent errors.'.

---

### Decision · Program_Chairs · 2024-09-26

**Decision:**

Accept (Poster)

**Comment:**

Since both AC and SAC are non-responsive, PCs decide to give accept based on current reviewers' comments.